# Impact of STAT Proteins in Tumor Progress and Therapy Resistance in Advanced and Metastasized Prostate Cancer

**DOI:** 10.3390/cancers13194854

**Published:** 2021-09-28

**Authors:** Celina Ebersbach, Alicia-Marie K. Beier, Christian Thomas, Holger H. H. Erb

**Affiliations:** 1Department of Urology, Technische Universität Dresden, 01307 Dresden, Germany; Celina.Ebersbach@uniklinikum-dresden.de (C.E.); AliciaMarie.Beier@uniklinikum-dresden.de (A.-M.K.B.); Christian.thomas@uniklinikum-dresden.de (C.T.); 2Mildred Scheel Early Career Center, Department of Urology, Medical Faculty and University Hospital Carl Gustav Carus, Technische Universität Dresden, 01307 Dresden, Germany

**Keywords:** STAT, prostate cancer, metastasis

## Abstract

**Simple Summary:**

Prostate cancer (PCa) is the second most common cancer in men and one of the leading causes of death. Signal transducers and activators of transcription (STATs) are transcription factors involved in the development and progression of several cancers, including PCa. STATs play a crucial role in the therapeutic resistance of prostate cancer, including antiandrogens and chemotherapy. They are further involved in metastatic PCa and are associated with advanced and high-grade PCa. To this day, inhibitors of STATs and their signaling molecules were tested in PCa but failed to succeed in clinical trials. This review discusses different functions of STATs in PCa and the current state of STAT-inhibitors in clinical trials.

**Abstract:**

Signal transducers and activators of transcription (STATs) are a family of transcription factors involved in several biological processes such as immune response, cell survival, and cell growth. However, they have also been implicated in the development and progression of several cancers, including prostate cancer (PCa). Although the members of the STAT protein family are structurally similar, they convey different functions in PCa. STAT1, STAT3, and STAT5 are associated with therapy resistance. STAT1 and STAT3 are involved in docetaxel resistance, while STAT3 and STAT5 are involved in antiandrogen resistance. Expression of STAT3 and STAT5 is increased in PCa metastases, and together with STAT6, they play a crucial role in PCa metastasis. Further, expression of STAT3, STAT5, and STAT6 was elevated in advanced and high-grade PCa. STAT2 and STAT4 are currently less researched in PCa. Since STATs are widely involved in PCa, they serve as potential therapeutic targets. Several inhibitors interfering with STATs signaling have been tested unsuccessfully in PCa clinical trials. This review focuses on the respective roles of the STAT family members in PCa, especially in metastatic disease and provides an overview of STAT-inhibitors evaluated in clinical trials.

## 1. Introduction

The protein family “signal transducers and activators of transcription” (STATs) are transcription factors first described at the beginning of the 1990s while investigating the cytokine signaling pathways [1,2]. The protein family comprehends STAT1, STAT2, STAT3, STAT4, STAT5a, STAT5b, and STAT6, encoded by different genes on different chromosomes [3]. These STAT proteins are involved in multiple biological processes such as immune response, mitogenesis, wound healing, cell survival, and cell growth by transmitting signals from the cell membrane into the nucleus [3,4]. The STAT proteins bind to specific response elements on the DNA in the nucleus, thereby inducing gene transcription. Based on their various functions, STAT proteins are essential in several health conditions such as autoimmune diseases and cancer [5,6]. Despite their broad spectrum of activity, only STAT3 affects embryonic development, as shown in STAT3 knock-out mouse experiments [7]. Different groups of signal proteins, including interleukins (e.g., IL-4 and IL-6), interferons (e.g., IFN-α/β), growth factors (e.g., EGF), and proteohormones (e.g., prolactin) activate STAT proteins [3]. 

In this review, we focus on the role of STAT proteins in metastatic prostate cancer (PCa) and summarize the current state of research regarding their involvement in metastatic castration-sensitive and castration-resistant PCa.

## 2. Prostate Cancer

PCa is the second most common cancer in men, with an estimated number of over 1.4 million new cases worldwide in 2020 and it is one of the leading causes of death in men, with an estimated 375,000 cancer-related deaths yearly [8]. PCa is androgen-dependent, requiring testosterone and its metabolite dihydrotestosterone (DHT) for growth and development. Consequently, androgen withdrawal causes reduced cell growth and induces apoptosis [9,10]. For locally confined PCa, radical prostatectomy and external beam radiotherapy are the treatment options with curative intent. However, in the setting of metastatic disease, metastatic hormone-sensitive PCa first-line therapy includes androgen deprivation therapy (ADT), or ADT combined with an antiandrogen (e.g., apalutamide, darolutamide, and enzalutamide), taxane-based chemotherapy (e.g., docetaxel, cabazitaxel), or the CYP17A1 inhibitor abiraterone [11]. Regrettably, after a median response duration of 18 months, most tumors develop ADT resistance and progress to castration-resistant PCa (CRPC) [12]. Therefore, treatment for CRPC includes antiandrogens, taxane-based chemotherapy, and abiraterone [11]. Since 2020, olaparib, a poly (ADP-ribose) polymerase (PARP)-inhibitor, has been EMA approved for the treatment of *BRCA1* and *BRCA2* positive mCRPC [11]. Unfortunately, despite treatment, the median survival of CRPC patients is only 19 months in average post castration resistance due to tumor progression and development of therapy resistance [13]. Therefore, new therapeutic targets and strategies are necessary for mCRPC. 

## 3. Characterization of STAT Family of Proteins

### 3.1. Structure

STAT genes are located on three different chromosomes: *STAT1* and *STAT4* are clustered together on chromosome 2 (locus 2q32.2 and 2q32.2-2q32.3, respectively), whereas *STAT2* and *STAT6* are located together on chromosome 12 (locus 12q13.13 and 12q13, respectively). *STAT3*, *STAT5a*, and *STAT5b* are clustered together on chromosome 17 (locus 17q21, 17ql1.2 and 17ql1.2, respectively) [14]. STAT proteins share seven functional domains (Figure 1): the N-terminal domain (ND), the coiled-coil domain (CCD), the DNA binding domain (DBD), the linker domain (LD), the SRC homology 2 (SH2) domain, the phosphotyrosyl-tail (Y) segment, and the transactivation domain (TAD) [3].

The ND mediates protein–protein interactions, like dimerization and tetramerization, and nuclear import of STAT proteins [3]. The CCD is involved in the nuclear translocation processes and interactions with regulatory proteins such as IRF9, c-JUN, and SMRT [3,15,16,17]. The DBD is highly conserved in all STAT proteins [18,19]. The domain facilitates the association of STAT proteins with palindromic DNA-binding sites, primarily the “Interferon-Gamma Activated Sequence “(GAS) and the “Interferon-Stimulated Response Element “(ISRE), located in the promoter region of the target genes. Just as the ND and CCD, the DBD is involved in nuclear import and export of STAT [3]. The LD affects mainly the transcriptional activity of STAT proteins. However, the domain is also involved in the DNA-binding and nuclear export processes. The SH2 domain is essential for the binding of STATs to their membrane-bound receptors. This domain is phosphorylated at its tyrosine residues mediating protein dimerization [3,20]. Moreover, it is also involved in the nuclear export processes of STAT proteins. The Y segment is the smallest domain of STAT proteins and has a specific tyrosine motif for every member of the STAT protein family [3]. The tyrosine residues point outward in folded STAT proteins and can be phosphorylated, enabling STAT proteins to dimerize by their SH2 domains. The C-terminal TAD plays an essential role in mediating the transcriptional activity of STAT proteins. It is variable in its sequence and length and holds various conserved serine-phosphorylation sites, mandatory for the interaction with different coactivators [3]. Furthermore, it has a central role in the ubiquitin-mediated degradation of STAT proteins [21]. 

For STAT proteins, different isoforms are known, which are due to alternative mRNA splicing or post-translational proteolytic processing [3]. The full-length STATs are α isoforms while the other shorter isoforms have been termed β, γ, or δ. The β isoforms still retain the tyrosine residue in the Y segment, Y701 for STAT1, Y690 for STAT2, Y705 for STAT3, Y693 for STAT4, Y694 for STAT5a, Y699 for STAT5b, and Y641 for STAT6. STAT2, STAT5a, and STAT5b do not contain any isoforms. STAT2 and all isoforms of STAT6 do not have a serine residue on their TAD domain. STAT1 α and β, and STAT3α possess S727 as serine residue. For STAT4 α, it is S721, for STAT5a, it is S726/S780, and for STAT5b, it is S731 [3]. 

### 3.2. STAT-Signaling

The canonical JAK/STAT signaling pathway (Figure 2A) is mediated by the intracellular Janus kinases (JAK) [3]. In its inactivated state, STAT proteins are localized in the cytoplasm. The JAK proteins are associated with different transmembrane receptors. Extracellular ligand binding to these transmembrane receptors activates the JAKs leading to tyrosine residues autophosphorylation of activated JAK proteins and phosphorylation of tyrosine residues on the cytoplasmatic domain of their transmembrane receptors (Figure 2A). Those phospho-tyrosine residues serve as binding sites for the SH2 domain of the STAT proteins. Bound STAT proteins are subsequently activated by the JAK proteins’ phosphorylation of tyrosine residues in the STAT’s Y segment. After phosphorylation, STAT proteins dissociate from the receptor, homo- or heterodimerize with other STAT proteins, and translocate into the nucleus. Nuclear STAT complexes bind to the GAS or ISRE elements of the DNA, recruit comodulators and RNA polymerase II, triggering their target genes expression [3,22]. 

The non-canonical JAK/STAT signaling pathway involves activation of the STAT proteins via intrinsic tyrosine kinase activity, such as the EGF receptor, or activation by oncoproteins such as v-src or v-Sis (Figure 2) [23,24]. 

Several mechanisms have been developed to negatively regulate the JAK/STAT pathway to prevent hyperactivity of the STAT transcription factors (Figure 2B) [3]. The “Suppressor of cytokine signaling” (SOCS) proteins are crucial in negatively regulating the JAK/STAT pathway, whereby different mechanisms have been described [25,26]. For example, SOCS1 binds directly to JAK proteins and inhibits their catalytic activity [27]. In contrast, SOCS3 binds to JAK-proximal sites on transmembrane receptors, inhibiting JAK activity and simultaneously preventing STAT from binding to these receptors [27]. 

Another protein family involved in the negative regulation of the JAKT/STAT signaling pathway is the “Protein inhibitors of activated STAT” (PIAS) protein family [28]. PIAS proteins regulate the JAK/STAT signaling pathway by direct interaction of STAT proteins, recruitment of histone deacetylases to inhibit transcription, and modulation of the transcriptional activity of STAT proteins by SUMOylation [29,30]. Besides the direct inhibition of the transcriptional activity of STAT proteins, PIAS proteins prevent the DNA-binding of STAT proteins to the GAS and ISRE motives.

## 4. STAT Family Members in Prostate Cancer 

Several studies have revealed that the STAT protein family is involved in therapy resistance and promotes tumor progression in several cancers, including PCa [6]. As STAT proteins are transcription factors, their primary regulation mechanism is the regulation of gene transcription (e.g., Bcl-xl, Mcl-1, c-Myc, Cyclin D1) [31]. These STAT-dependent genes are involved in pro-survival, anti-apoptotic, and pro-proliferation signaling pathways, metastasis, and epithelial-mesenchymal transition, as discussed later below. Especially the STAT family members STAT1, 3, 5a, 5b, and 6 are essential for tumor progression and development of therapy resistance (Table 1) in advanced PCa. Until now, the STAT family members STAT2 and STAT4 are less researched in PCa.

### 4.1. Impact of STAT Proteins on Other Signaling Pathways in Prostate Cancer

Next to their function as a transcription factor, STAT proteins have been reported to regulate other signal pathways involved in PCa survival and progression. The nuclear factor kappa-light-chain-enhancer of activated B cells (NF-κB) has been frequently demonstrated to be constitutively active in PCa [48]. The transcription factor regulates cell survival, tumor invasion, metastasis, and chemoresistance. Acetylated STAT1 has been reported to influence the transcriptional activity of NF-κB by direct interaction [49]. Moreover, several studies have identified versatile and antagonistic interactions between NF-κB and STAT3, promoting colon, gastric, and liver cancers [50]. This interaction also regulates the dialog between the cancer cells and the tumor microenvironment, especially with infiltrating immune cells. In the PCa cell line Du145, STAT3 is mandatory to maintain NF-κB and its anti-apoptotic effects constitutively active [51]. Conversely, NFKB regulates the activity of STAT3 by regulating the transcription of IL-6 [52]. High STAT3 and NF-κB activity have been linked to PCa progress and development of metastasis [53,54]. Nonphosphorylated STAT3 has been revealed to bind to the NF-κB and thus facilitating its activation independently of IKK activity and prolong the presence of active NF-κB dimers in the nucleus [50]. Active NF-κB has been reported to induce androgen receptor splicing variants (ARVs) expression. These ARVs lack the ligand-binding domain and are constitutively active [55]. These splicing variants are associated with castration-resistant prostate cancer, resistance to second-generation antiandrogens, and metastasis [56,57,58,59]. Moreover, NF-κB transactivity in PCa increases osteoclastogenesis by up-regulating osteoclastogenic genes, thereby contributing to bone metastasis [60].

Next to NF-κB, STAT proteins have been shown to interact with different steroid hormone receptors such as the androgen receptor and the glucocorticoid receptor [35,40,41,61,62]. STAT proteins thereby regulate localization, stabilization, and DNA binding of these transcription factors and subsequently their transactivity. The AR is a key player in PCa progression and the high AR transactivity facilitates the metastatic process and maintenance of metastatic disease [63,64]. 

Besides transcription factors, STAT proteins have also been reported to interact with the PTEN/PI3K-AKT signal pathway [65]. AKT, also known as protein kinase B, promotes prostate tumor growth and metastasis [66,67]. The interaction between STAT3 and AKT signaling pathways results in invasive carcinoma development in mice [68]. Similar results were shown in mouse and patient-derived xenografts with increased expression of IL-6 in combination with PTEN loss, which results in constitutively active AKT, enhanced expression of STAT3, and promoted tumor progression [69]. Contrary to that, loss of STAT3 in *PTEN*-deficient mice enhanced tumor growth and metastasis [70]. In addition, crosstalk between STAT1 and AKT is involved in the expression of *clusterin*, which is a driver in docetaxel resistance [44,71].

Furthermore, interactions of STAT proteins with the programmed death receptor 1(PD-1)/programmed death-ligand 1(PD-L1) immune checkpoint are involved in PCa progression and resistance. In vitro and in vivo experiments showed that IL-6 promotes resistance of CRPC cells to natural killer (NK) cell-mediated cytotoxicity via STAT3 and upregulation of PD-L1. Therefore, an enhanced effect of combined PD-L1 and JAK/STAT3 inhibition on NK cell-mediated cytotoxicity in CRPC is postulated [72].

### 4.2. Mutations in STAT Proteins in Prostate Cancer

Gene mutations can lead to changes in protein function and activity, thus influencing disease progression. In addition, especially accumulations in oncogenic and survival pathways can lead to tumor progress and development of therapy resistance [73,74]. Mutations in transcription factors can lead to several diseases and include both gain and loss of function mutations [75]. 

Analysis of public data sets revealed that although STAT proteins are involved in cancer progression and survival, mutation frequencies (≤1.1%) are negligible in metastatic PCa (Figure 3). Most mutations are found in the STAT proteins STAT1, STAT3, STAT5a, and STAT5b. None of the mutations have been identified as cancer hotspots in metastatic PCa. Furthermore, the here presented mutations are not described in prostate cancer or other malignancies and therefore seem insignificant in tumorigenesis and progression in general. Therefore, STAT mutations don’t seem to be the main driver for high STAT activity in PCa.

### 4.3. STAT1

STAT1 was the first described STAT in 1989 and is the main mediator of interferon-alpha and beta signaling [103]. The *STAT1* gene encodes two STAT1 isoforms: the transcriptionally active STAT1α and the dominant-negative inhibitor STAT1β [104].

In PCa and other tumor entities, STAT1 acts as a tumor suppressor and oncogene [105]. Especially in the PCa early stages, STAT1 is assumed to function as a tumor suppressor [106]. Elevated STAT1 expression correlates with prolonged cancer-specific survival in patients with localized PCa, while no STAT1 expression in patients with advanced PCa is accompanied by early biochemical recurrence [107]. Moreover, STAT1 is elaborated in PCa therapy resistance. For example, increased STAT1 expression is found in docetaxel-resistant PCa cells [44,45]. In the metastatic PCa cell line, Du145 STAT1 expression and activation were increased after long-term docetaxel treatment, causing an increased *clusterin* expression. The increased *clusterin* expression mediates anti-apoptotic effects and inhibits docetaxel-induced apoptosis [44]. Furthermore, a connection between increased STAT1 expression and radio-resistant tumor cells was reported in head and neck squamous cell carcinoma [108,109]. Irradiation activates the STAT1 signaling pathway causing cytoprotective effects against radiation therapy [109]. Additional studies confirmed the effects of radiation on STAT1-expression in breast cancer, gliosarcoma, and metastatic PCa cell lines, where STAT1 was significantly up-regulated after radiation, indicating a universal role of STAT1 in radioresistance, including PCa [32]. Preclinical studies confirmed that IFN/STAT1 pathway mediates radioresistance. It is suggested that STAT1 creates a radioprotective tumor microenvironment by shielding the immune responses and activating survival signaling pathways [110]. However, the exact mechanisms by which STAT1 might contribute to radioresistance are still not completely revealed. 

In other tumor entities, STAT1 is also described as a double-edged sword. For example, high osteopontin levels in bladder cancer promote invasion and metastasis by activating JAK1/STAT1 signaling [111]. In other cancers, STAT1 emerges as a suppressor of metastasis. For example, the phosphatase Myotubularin-related protein 2 (MTMR2) promotes invasion and metastasis in gastric cancer by inactivating IFNγ/STAT1 signaling pathway [112]. In vivo studies on breast cancer revealed anti-metastatic effects of STAT1 by showing that STAT1^−/−^-mice displayed more lung metastases than wild-type mice [113]. Similar results were demonstrated in head and neck squamous cell carcinoma (HNSCC), where STAT1^−/−^-mice showed more metastasis than STAT1^+/+^-mice [114]. However, little is known about the role of STAT1 in the development of PCa metastasis so far.

As the exact role of STAT1 as a tumor suppressor or oncogene is not clarified yet, more investigations are mandatory to identify STAT1 as a therapeutic target.

### 4.4. STAT2

STAT2 was described as a co-factor only involved in type I interferon signaling. The heterodimer STAT1:STAT2 is a key complex in interferon-alpha signaling and is involved in the immune response to viral infections [103]. Currently, limited information is available on STAT2 in PCa. Knockdown of STAT2 has been linked to decreased cell proliferation, migration, and invasion in the metastatic PCa cell line model PC-3 [115]. It is suggested that STAT2 acts via the TRIM66-STAT2-IL-2-axis and has an oncogenic role in PCa [115]. 

Moreover, in other tumor entities, even less is known about the role of STAT2. For example, high expression of STAT2 was associated with poor prognosis in ovarian cancer [116]. Furthermore, in cervical cancer and premalignant cervical-intraepithelial neoplasia (CIN), STAT2 expression was increased compared to benign cervicitis [117]. STAT2 also seems to be involved in colorectal carcinogenesis, where STAT2^−/−^-mice showed fewer and smaller tumors than wild-type mice. Similar results were demonstrated regarding skin carcinogenesis in STAT2^−/−^-mice which developed papilloma later than wild-type mice, whereas cancer incidence did not differ between the two groups [118].

STAT2 still remains nearly unresearched in PCa, requiring further investigations.

### 4.5. STAT3

The transcription factor STAT3 was first described in 1994 in IL-6–stimulated hepatocytes [119]. STAT3 isoforms STAT3α and STAT3β are co-expressed with a generally higher expression of STAT3α [120]. STAT3 β is generally known as a dominant-negative regulator of transcription [120]. The STATα isoform interacts with protein kinase Cε and promotes cancer cell invasion in prostate cancer [121].

STAT3 is the most investigated STAT protein in PCa and is involved in many oncogenic pathways while aberrantly activated in ~50% PCa patients [122]. Moreover, the transcription factor is involved in the regulation of crucial effectors in survival (e.g., Bcl-2, Bcl-xL, Mcl-1), proliferation (e.g., Cyclin D1, D2, c-Myc), and metastasis (e.g., Twist, MMP-2, 9, 7) [64,123]. Besides, it acts as a repressor of tumor suppressor genes such as *TP53*, as shown in vitro using the cell line 3T3 [124]. The study revealed that blocking STAT3 increases the expression of p53, leading to p53-mediated tumor cell apoptosis.

In primary PCa, STAT3 is constitutively active as immunohistochemical analysis of primary PCa tissue revealed, while elevated levels of phosphorylated STAT3 correlate with higher Gleason scores [122]. Furthermore, STAT3 inhibition by small molecules such as Galiellalactone causes apoptosis-mediated tumor regression in vitro and reduces regional and distal lymph nodes metastases in vivo [122,125,126]. Moreover, in vitro experiments revealed that Galiellalactone reduces cell viability and invasion and induces apoptosis in metastatic PCa cell line DU145 [126]. 

Advanced PCa and PCa metastases display high STAT3 expression and activity [127]. In the cancer-bone microenvironment, STAT3 signaling is essential for cell migration and viability [128]. STAT3 expression was elevated in bone metastases compared to lymph node and visceral metastases, indicating the STAT3 signaling pathway is critical in bone metastasis [127]. Contrary to that, loss of STAT3 in *PTEN*-deficient mice enhanced tumor growth and metastasis, suggesting that STAT3 suppresses oncogenic progression in *PTEN* -deficient PCa [70]. Although, in this context, the differing anatomy and histology of the murine prostate have to be taken into account [129]. 

Constitutively active STAT3 is often observed at the invasive front of human tumors adjacent to inflammatory cells and drives tumor growth and metastasis even under antiandrogen treatment [36,37,130]. It is suggested that the interaction of STAT3 with the N-terminal domain of the androgen receptor (AR) leads to a change in androgen sensitivity and increased AR-activation and supports the progress to antiandrogen resistance [36,37]. In vitro experiments validated the role of constitutively active STAT3 in antiandrogen resistance by showing that inhibition of STAT3-activity leads to a reversal of enzalutamide-resistance [38,39]. 

Additionally, STAT3 is involved in neuroendocrine differentiation, promoting resistance to androgen deprivation therapy and taxane-containing chemotherapy, displaying a more aggressive and lethal PCa phenotype [34]. Involvement of STAT3 in docetaxel resistance was also found in the stress and survival pathway NF-κB [46]. Increased STAT3 expression and activity in docetaxel-resistant cell models lead to an increased PIM1-expression causing activation of NF-κB signaling pathway, which mediates resistance to docetaxel by inhibiting docetaxel-induced apoptosis [46,47]. 

The intense investigations have identified multiple roles of STAT3 in cancer initiation and progression. Therefore, STAT3 seems to be a valid target for new therapeutic strategies.

### 4.6. STAT4

The STAT4 protein is involved in the development and maturation of Th1 cells [131]. STAT4 has two isoforms, STAT4α and STAT4β, generally expressed at lower levels [132]. IL-12 activates STAT4β for a more extended period than STAT4α [133]. In PCa, activated STAT4 is present, although not statistically elevated compared to normal prostate tissue [134]. Besides that, the role of STAT4 is not well investigated in PCa yet. STAT4 seems to operate as a tumor suppressor in hepatocellular carcinoma (HCC). Here, an increased STAT4 expression correlates with better recurrence-free survival, whereas downregulation of STAT4 in HCC showed more aggressive tumors, enhanced cell proliferation, and worse clinical outcomes [135,136]. In patients with gastric cancer, high expression of STAT4 indicates better disease-free survival, and STAT4 expression being a significant factor in tumor recurrence [137]. In vivo studies regarding HNSCC showed a higher rate of metastasis in STAT4^−/−^- mice indicating a role of STAT4 in preventing tumor metastasis in HNSCC. In contrast to the involvement in metastasis, primary tumor development was not affected [138]. Contrarily, STAT4 plays a role in tumor progression in colorectal cancer (CRC). In CRC, high STAT4 expression correlates with increased invasiveness. At the same time, inhibition of STAT4 repressed growth and invasion of CRC cell lines [139]. STAT4 regulates complement factor H (CFH) in lung cancer, which inhibits the complement system and mediates resistance to therapy with monoclonal antibodies, which generally enhances complement activation leading to cellular cytotoxicity. STAT4 overexpression in lung cancer leads to overexpression of CFH, which then mediates inadequate therapy response [140]. Moreover, overexpression of activated STAT4 was shown in epithelial cells of ovarian cancer, which correlated with poor clinical outcome and progress to metastasis by a different mechanism, including EMT and interaction with tumor stroma [141].

STAT4 has been influential in other tumor entities, but remains nearly unresearched in PCa, requiring further investigations.

### 4.7. STAT5

STAT5, which refers to the related proteins STAT5a and STAT5b, plays an essential role in the progression of PCa to CRPC [142,143,144,145]. The transcription factors STAT5a and STAT5b are discretely encoded and mediate signals for a broad spectrum of cytokines. In particular, STAT5 mediates the early progress of prolactin-driven prostate tumorigenesis [146]. STAT5 and its prolactin-driven effects were initially shown in lactating sheep, bovine, and rodents as so-called mammary gland factor (MGF). Subsequently, they were assigned to the STAT family of proteins [147,148,149]. In PCa tissue, STAT5 expression correlates with high Gleason scores and is predictive for an early recurrence of PCa after radical prostatectomy [150,151]. Additionally, androgen withdrawal by ADT leads to an increased STAT5 expression in PCa tissue [40]. A *STAT5* gene amplification accompanies the high STAT5 expression during the progression of PCa to CRPC [152]. 

In human PCa tissue, STAT5a/b is more frequently active in primary PCa during ADT and is also active in 95% of hormone-refractory PCa specimens with ADT [35]. In addition, STAT5 increases the transcriptional activity of the AR by influencing protein stability in PCa cells in vivo and in vitro [35,40]. Furthermore, active STAT5a/b protects antiandrogen-liganded AR from proteasomal degradation, while STAT5a/b knockdown combined with antiandrogen treatment enhances proteasomal degradation of AR followed by suppression of tumor growth [41]. 

A positive JAK2/STAT5 feed-forward loop induced by the second-generation antiandrogen enzalutamide was reported. Enzalutamide-liganded AR activates JAK2, leading to phosphorylation of STAT5, increasing the mRNA and protein levels of JAK2 [42]. Inhibition of STAT5 by IST5-002, a specific STAT5-inhibitor, combined with enzalutamide, suppressed the growth of CRPC in vivo more efficiently than enzalutamide alone [42]. Concurrently, siRNA-mediated knockdown of STAT5b led to a resensitization of enzalutamide-resistant cell line MR49F to enzalutamide treatment [43]. STAT5 is also required for Rad51 expression, which plays a crucial role in homologous recombination DNA repair. Genetic and pharmacologic inhibition of STAT5 sensitized PCa cells in vitro and in vivo to radiation therapy [33]. 

In addition, PCa xenotransplantation studies revealed an essential role of STAT5 in tumor initiation and progression. Increased STAT5 expression correlates with a mesenchymal phenotype caused by STAT5-induced epithelial-mesenchymal transition (EMT) [40,45,153]. EMT is a complex process where endothelial cells lose their endothelial markers and gain mesenchymal features, allowing them to migrate. EMT in tumors, therefore, promotes angiogenesis and metastasis [154]. In 61% of clinical PCa metastases, active STAT5 was found, while the number of lung metastases was 11-fold higher in mice expressing active STAT5 [155]. Thus, increased metastasis in PCa expressing high levels of STAT5 may be caused by STAT5-induced EMT [153]. EMT-induced phenotypes have also been associated with resistance to chemotherapy, especially to docetaxel [13,45,156]. Increased STAT5 levels have already been identified in docetaxel-resistant mCRPC cell line models. However, STAT5’s role in docetaxel resistance is still unknown [157]. 

Next to STAT3, STAT5 has been highly investigated in PCa, and its role in tumor progression has been described. Therefore, it is a valid target for new therapeutic strategies. 

### 4.8. STAT6

The transcription factor STAT6 was first discovered in 1988 and is a significant player in developing T-helper type 2 (Th2) cells and Th2 immune response and is mainly activated by IL-4 and IL-13 [158,159]. Three isoforms of STAT6 are known, termed STAT6a, STAT6b, and STAT6c. In human tissue, STAT6a expression was 2–4 times higher than STAT6b and 2.7–13.8 times higher than STAT6c. Further, the expression of STAT6a and STAT6b enhances, whereas STAT6c inhibits IL-4-mediated DNA synthesis in vitro [160]. In PCa, tissue microarray, Western blot, and electromobility shift assay analysis revealed that STAT6 activity and expression are elevated compared to benign prostate [134,161]. Moreover, tissue analysis of PCa and normal prostate linked STAT6 expression to high-grade PCa tissues (Gleason score ≥ 4 + 3) and larger tumor size [162,163]. STAT6 expression is increased in fibromuscular stroma regions of PCa [163]. 

in vitro studies revealed a possible role of STAT6 in cell survival as siRNA mediated STAT6-knockdown-induced apoptosis in metastatic PCa cell line DU145. Moreover, the STAT6 knockdown could be linked to the decreased ability of PCa cell lines to migrate, an essential step in tumor migration and metastasis [163]. Furthermore, IL-4 increased the clonogenic potential of primary PCa cell lines, which could be reversed by treatment with selective STAT6-inhibitor AS1517499 [164], indicating IL-4/STAT6 involvement in the clonogenic ability of PCa cells [161]. Additionally, several miRNAs were found acting as tumor suppressors while suppressing tumor growth, inducing apoptosis, or inhibiting metastasis in PCa by targeting STAT6, thereby underlining the cancer-promoting functioning of STAT6 in PCa [165,166,167]. Eventually, the transcriptional activity of STAT6 is up-regulated by AnxA2, which itself is suggested to be involved in PCa metastasis [168].

Current data describe STAT6 as an essential factor in metastasis in PCa, but its exact role in tumor progression is still largely unknown, and further investigation is needed.

### 4.9. Role of STAT Proteins in Prostate Cancer Stem Cells

Cancer stem cells (CSCs) are small subpopulations of cancer cells that share similar stem or progenitor cells [169,170,171,172]. These characteristics include the ability to self-renewal and multi-lineage differentiation, which drive tumor growth and progression. Moreover, it is suggested that CSC are responsible for tumor heterogeneity. In PCa, the existence is still controversially discussed. The central hypothesis is that PCa develops from luminal cells; however, there is growing evidence that PCa arises from undifferentiated basal cells [173,174,175]. STAT proteins have been identified to be implicated in stem cell-like PCa cells [176,177]. STAT3 has been recognized by gene expression profiling of PCa CSCs to regulate the stem cell niche and tumor initiation [177,178,179]. Moreover, inhibition of STAT3 by the galiellalactone effectively reduced PCa stem-like cell population [180]. The IL-4-activated STAT6 has been reported to increase the clonogenic potential of prostate stem-like cells, and therefore IL-4 provides a favorable niche for clonogenic growth [161]. In other tumor entities, the JAK-STAT pathway has been identified to regulate sphere-forming capacity, efflux activity, ROS depletion, and resistance mechanisms to chemotherapy and radiotherapy [181,182,183]. Therefore, STAT protein in CSC seems to be a valid target to reduce the tumor driving cell population. 

### 4.10. STAT-Inhibitors in PCa in Clinical Trials

Since STAT proteins, especially STAT3 and STAT5, emerge as key players in PCa development and progression, several inhibitors have been developed and tested in clinical trials (Table 2).

Two studies with the JAK1/2-Inhibitor Ruxolitinib (INCB018424), subsequently inhibiting STAT3 and STAT5, in PCa had been planned. However, a phase II trial on treating patients with mCRPC with orally applied Ruxolitinib was terminated due to less than 2 of the first 22 patients showing a PSA50 response (NCT00638378). In addition, another study aiming to assess the effect of Ruxolitinib on tumor-infiltrating myeloid cells and additional immune subsets in PCa patients was stopped early because of recruitment difficulties due to eligibility criteria (NCT03274778). AZD1480 is another inhibitor of JAK1 and JAK2, which effectively inhibited IL-6/STAT3-driven metastasis in vivo [184] and was investigated in a phase I study to assess the safety and tolerability of AZD1480 in 38 patients with advanced solid malignancies, including PCa. However, the study was terminated due to dose-limiting toxicities [185]. 

Siltuximab (CNTO 328) is a chimeric monoclonal antibody against IL-6 interfering with the IL-6/JAK/STAT3 signaling pathway. In a phase I trial, 15 patients were treated preoperatively with different doses of siltuximab. Treated patients showed no drug-related adverse effects and higher levels of apoptosis in comparison with the control group (5 patients) [186]. A phase II clinical trial of siltuximab in 53 patients with CRPC, who were treated with one prior chemotherapy, showed no significant activity as monotherapy in this subset of patients by presenting a prostate-specific antigen (PSA) response rate of only 3.8% and progression-free survival of fewer than 7 weeks [187]. A randomized phase II trial in patients with mCRPC with prior docetaxel-based chemotherapy compared treatment with siltuximab combined with mitoxantrone/prednisone (48 patients) to mitoxantrone/prednisone alone (49 patients). No improvement in clinical outcome was demonstrated by treating patients in combination with siltuximab as median overall survival was 394 days with mitoxantrone/prednisone and 311 days with siltuximab plus mitoxantrone/prednisone (hazard ratio = 1.45, *p* = 0.226) [188]. Combination therapy of siltuximab with docetaxel was tested in 39 patients with mCRPC in a phase I trial. The study aiming to test the safety of combined treatment with siltuximab and docetaxel was completed. Siltuximab did not affect docetaxel’s pharmacokinetics, but the combination therapy did cause treatment-limiting adverse effects [189]. 

Tocilizumab is a monoclonal anti-IL6-receptor-antibody. A phase II study of neoadjuvant atezolizumab (a monoclonal PD-L1-antibody)-based combination therapy with tocilizumab in patients with localized PCa before radical prostatectomy is currently recruiting (estimated enrollment: 68 patients) and about to be completed in 2022 (NCT03821246). 

Pacritinib (SB1518) is an inhibitor of JAK2 and FTL3, which will be investigated in a phase II study in patients with histologically confirmed prostate adenocarcinoma with prior radical prostatectomy or definitive radiation. The study started/began in June 2021 and is currently recruiting (estimated enrollment: 46 patients) and will be completed in 2026 (NCT04635059). 

Furthermore, when basic and translational research data are promising, most clinical trials failed or showed no benefit. Therefore, further analysis of these trials is mandatory to identify the reason for the failure. 

## 5. Future Directions

The role of STAT proteins has been investigated intensely in PCa and other tumor entities. Especially, basic research has revealed the involvement of the STAT pathways in multiple areas such as tumor progression, therapy resistance, and metastasis. However, translation of these results seems to be challenging, and they result in failed clinical trials. As most results have been obtained from mono cell culture experiments, a possible reason for the failing translation may be the missing tumor microenvironment and immune cell interaction. Therefore, the findings obtained from basic research need to be validated in future experiments on near-patient models such as organoids, co-culture systems, patient-derived xenografts, or ex vivo tissue slices.

## 6. Conclusions

The biological background and the driving forces behind metastasized PCa are still largely unknown. STAT proteins have been identified as critical players in the tumor biology of several entities. Moreover, in PCa, STAT proteins have been involved in several tumor biological processes, especially in advanced and metastasized stages. STAT1 emerges as a tumor suppressor in the early stages of PCa. However, STAT1 seems to be involved in radio- and docetaxel-resistance representing a different role in progressed PCa. STAT2 and STAT4 remain nearly unresearched in PCa but could potentially be of interest as studies in other malignancies showed their involvement in cancer progression as well as suppression. STAT3 is the best-researched STAT protein in PCa. It emerges as a critical player in progression to mCRPC, especially in bone metastasis and therapy resistance to antiandrogens and docetaxel. This fact makes it a viable therapeutic target for which different inhibitors have already been developed but, unfortunately, to date, failed to show success in clinical trials. STAT5a and STAT5b play a crucial role in PCa progression and interfere with different therapy options and resistances. They are involved in EMT and metastatic progression, thus displaying a potential therapy target, yet only a few promising inhibitors have been developed until today. Eventually, STAT6 acts as a pre-cancerous factor in PCa, promoting progression to metastatic disease. In general, STAT proteins provide potential biomarkers and viable therapeutic targets in advanced and metastasized PCa. However, until now, clinical trials targeting STATs in PCa failed, and therefore, more investigations about the function of STAT proteins in PCa are mandatory.

## Figures and Tables

**Figure 1 cancers-13-04854-f001:**
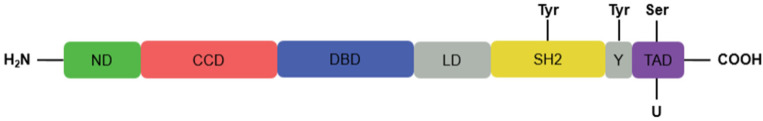
Schematic illustration of STAT domains. STAT proteins consist of seven domains: N-terminal domain (ND), coiled-coil domain (CCD), DNA binding domain (DBD), linker domain (LD), SRC homology 2 (SH2) domain, phosphotyrosyl-tail (Y) segment, and transactivation domain (TAD). In addition, phosphorylation sites are provided as tyrosine (Tyr) residues on the SH2 domain and Y segment and as serine (Ser) residues on the TAD domain. TAD domain is also involved in ubiquitination (U).

**Figure 2 cancers-13-04854-f002:**
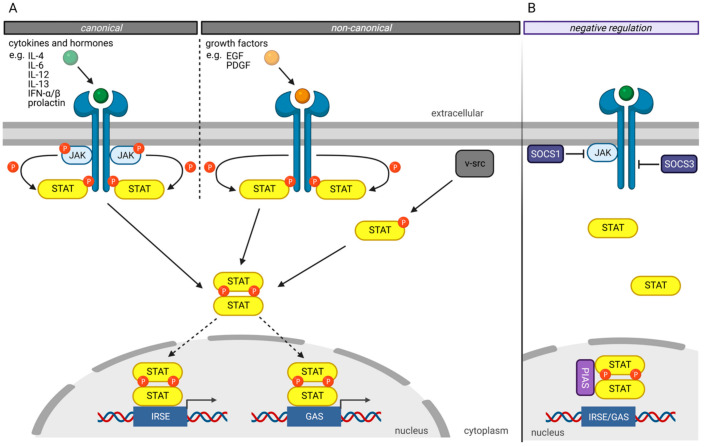
Schematic illustration of different ways of STAT activation and negative regulation. (**A**) STATs are activated by phosphorylation. In the canonical JAK/STAT pathway, extracellular binding of cytokines or hormones to cytokine receptors activates receptor-associated JAKs, which then phosphorylate STATs. Non-canonical signaling includes growth hormone receptors, like the EGF-receptor, which can phosphorylate STATs directly. Furthermore, viral oncoproteins, like v-src, activate STAT constitutively. Phosphorylated STATs form homo- or heterodimers and translocate to the nucleus. They regulate gene expression by binding to IRSE or GAS in the promoter region of their target genes. (**B**) Negative regulation of JAK/STAT pathway includes inhibition of JAK proteins by SOCS1, whereas SOCS3 binds to JAK-proximal sites on transmembrane receptors and inhibits binding of STATs to receptors. PIAS binds to STAT dimers in the nucleus, thereby inhibiting their transcriptional activity. (Created with BioRender.com, accessed on 10 September 2021).

**Figure 3 cancers-13-04854-f003:**
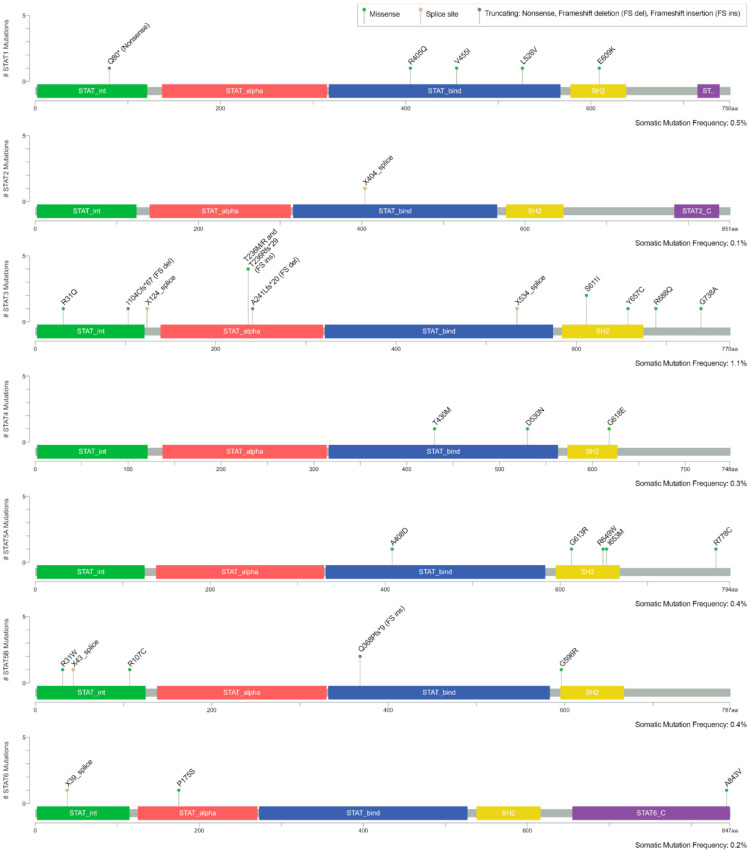
Mutations in STAT proteins in metastatic PCa [76,77]: This combined study contains 1562 metastatic samples from 1451 PCa patients in 16 studies: Metastatic Prostate Adenocarcinoma (MCTP, Nature 2012) [78], Metastatic Prostate Adenocarcinoma (SU2C/PCF Dream Team, PNAS 2019) [79], Metastatic Prostate Cancer (SU2C/PCF Dream Team, Cell 2015) [80], Metastatic castration-sensitive prostate cancer (MSK, Clin Cancer Res 2020) [81], Prostate Adenocarcinoma (Broad/Cornell, Cell 2013) [82], Prostate Adenocarcinoma (Broad/Cornell, Nat Genet 2012) [83], Prostate Adenocarcinoma (CPC-GENE, Nature 2017) [84], Prostate Adenocarcinoma (Fred Hutchinson CRC, Nat Med 2016) [85], Prostate Adenocarcinoma (MSK, Eur Urol 2020) [86], Prostate Adenocarcinoma (MSKCC, Cancer Cell 2010) [87], Prostate Adenocarcinoma (MSKCC, PNAS 2014) [88], Prostate Adenocarcinoma (MSKCC/DFCI, Nature Genetics 2018) [89], Prostate Adenocarcinoma (SMMU, Eur Urol 2017) [90], Prostate Adenocarcinoma (TCGA, Cell 2015) [91], Prostate Adenocarcinoma (TCGA, PanCancer Atlas) [92,93,94,95,96,97,98,99,100,101], Prostate Cancer (MSKCC, JCO Precis Oncol 2017) [102], generated on 10 March 2021.

**Table 1 cancers-13-04854-t001:** STAT proteins in therapy resistance in PCa.

Resistance	Therapy	Signaling	Experimental Setting	References
radioresistance	radiation	STAT1 STAT5	in vitro, in vivo in vitro, in vivo, ex vivo	[32] [33]
castration-resistance	ADT	STAT3 STAT5a/b	in vitro human PCa tissue (357 patients)	[34] [35]
Antiandrogen-resistance	antiandrogens like enzalutamide	STAT3 STAT5a+b	in vitro, in vivo in vitro, in vivo, ex vivo, human PCa tissue (132 and 20 patients, respectively)	[36,37,38,39] [35,40,41,42,43]
chemotherapy-resistance	chemotherapies like docetaxel	STAT1 STAT3	in vitro in vitro, 72 PCa patients (serum and tissue samples)	[44,45] [34,46,47]

**Table 2 cancers-13-04854-t002:** STAT-inhibitors in PCa in clinical trials. (N/A: not applicable).

Inhibitor	Target	Phase	Patient Number	Status/Results	Clinical Trials Identifier	References
Ruxolitinib (INCB018424)	JAK1/2	II	22 0	Terminated due to missing PSA50 response and recruitment difficulties, respectively	NCT00638378 NCT03274778	N/A N/A
AZD1480	JAK1/2	I	38	Terminated due to dose-limiting toxicities	NCT01112397	[185]
Siltuximab (CNTO 328)	IL-6	I II II I	20 53 97 39	No drug-related adverse effects, No activity as monotherapy in CRPC, No improvement in clinical outcome, No marks on docetaxel pharmacokinetics, Treatment-limiting adverse effects in combination with docetaxel	N/A NCT00433446 NCT00385827 NCT00401765	[186] [187] [188] [189]
Tocilizumab	IL-6-R	II	N/A	Currently recruiting, completed in 2022	NCT03821246	N/A
Pacritinib (SB1518)	JAK2 FTL3	II	N/A	Currently recruiting, completed in 2026	NCT04635059	N/A

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
