# Peer review of "Impact of STAT Proteins in Tumor Progress and Therapy Resistance in Advanced and Metastasized Prostate Cancer"

_cancers, 2021, doi:10.3390/cancers13194854_

Round 1

Reviewer 1 Report

The manuscript by Ebersbach et al. reviews the role of STAT family members in prostate cancer (PCa), particularly metastatic castration-sensitive and castration-resistant PCa, and provides an overview of STAT inhibitors and their current status in clinical trials.

The manuscript is well written and interesting. Through this manuscript, the reader will get a very good overview and a nice summary of the state of research on each STAT protein, STAT inhibitor, and clinical trials.

In the manuscript, however, a few small things should still be changed/discussed:

*Line 112/ Line 263: Please write the full name first before abbreviating GAS, IRSE, and CFH.

* Line 294: "HR” is an unnecessary abbreviation here, which otherwise does not occur in the test.

* Since STAT4 is involved in the development and maturation of Th1 cells and STAT6 is involved in the development of Th2 cells and the Th2 immune response, it would be of interest to discuss the effects or benefits of STAT4 or STAT6 inhibition in PCa, particularly with regard to the tumor microenvironment.

Author Response

On behalf of all authors, we would like to take this opportunity to express our sincere gratitude to the reviewers who identified areas of our manuscript that needed correction or modification. Their insightful comments have led to an improvement in our manuscript. Below you find the detailed response to the reviewers' comments:

*Line 112/ Line 263: Please write the full name first before abbreviating GAS, IRSE and CFH.

The abbreviations have been introduced in the text.

* Line 294: "HR" is an unnecessary abbreviation here, which otherwise does not occur in the test.

The abbreviation has been removed

* Since STAT4 is involved in the development and maturation of Th1 cells and STAT6 is involved in the development of Th2 cells and the Th2 immune response, it would be of interest to discuss the effects or benefits of STAT4 or STAT6 inhibition in PCa, particularly with regard to the tumor microenvironment.

We want to thank the reviewer for their comment. In this review, we tried to focus on the role of STAT proteins in PCa cells. The influence of STAT on the PCa microenvironment is fascinating; however, the data situation is so far very limited and the area under-researched. With the introduction of checkpoint inhibitors, the growing interest in immune cell infiltrates, and the knowledge about the microenvironment, we hope this situation will soon change.

Reviewer 2 Report

Inn present review, authors discusses the function of transcription factor STAT in prostate cancer and sheds light on its therapeutic importance. I have several reservations. My comments are appended as below: 1. In first place, authors should reframe the title; it should be conclusive, denoting summary of review in a line. 2. Introduction section: Prostate cancer is major player along with STAT3. Authors should first describe the pathology of prostate cancers, clinical issues like resistance and metastasis and then come on describing STAT3. 3. Characterization of STAT family of proteins: please indicate the genomic location. 4. Figure 1- please annotate the phosphorylation and ubiquitination sites. 5. Figure 2- please indicate the extracellular stimuli clearly. In addition, indicate the mediators of canonical and non-canonical signaling, SOCS and PIAS. 6. Line 127- please list out in table the therapy, resistance, signaling and reference. 7. Line 136- please explain in detail the role of STAT family of proteins in progression and metastasis along with the explicit signaling. 8. Table 4- Please include additional details as no of patients and the competent reference. 9. Authors should include a figure illustrating the role of STAT isoforms in prostate cancer. 10. While describing the trials in the text, please indicate the no of patients. 11. Line 221, 317- please describe in detail. What kind of in vitro study? Which cell lines/model system was used? 12. Line 323- indicate miRNAs. Ina addition, also evaluate the role of lncRNAs. 13. Discuss the role of STAT signaling in stem cells as the later are known for resistance and recurrence. 14. Please also discuss the cross talk of STAT with other pathways and NFKB contributing to prostate cancer metastasis and resistance. 15. Reference 69- please describe the mechanism in detail and this seems important study. 16. Authors should include ‘future directions’ section.

Author Response

Review 2

On behalf of all authors, we would like to take this opportunity to express our sincere gratitude to the reviewers who identified areas of our manuscript that needed correction or modification. Their insightful comments have led to an improvement in our manuscript. Below you find the detailed response to the reviewers' comments:

  1. In first place, authors should reframe the title; it should be conclusive, denoting summary of review in a line.

We want to thank the reviewer for his advice. We tried to change the title to be more informative about the content of the review.

  1. Introduction section: Prostate cancer is a major player along with STAT3. Authors should first describe the pathology of prostate cancers, clinical issues like resistance and metastasis and then come on describing STAT3.

We moved our information about prostate cancer to a separate section. Therefore, all the information about pathology, therapy, development of CRPC should now be more precise.

The authors discussed rearranging the STAT3 section. In our opinion, although STAT3 is the most studied STAT in PCa, this may change in the future. Therefore, we decided against weighing the importance of the STAT proteins and kept the order.

  1. Characterization of STAT family of proteins: please indicate the genomic location.

The genomic locations have been added.

  1. Figure 1- please annotate the phosphorylation and ubiquitination sites.

The phosphorylation and ubiquitination sites have been added.

  1. Figure 2- please indicate the extracellular stimuli clearly. In addition, indicate the mediators of canonical and non-canonical signaling, SOCS and PIAS.

The information has been added to the figure.

  1. Line 127- please list out in table the therapy, resistance, signaling and reference.

A table with the asked information has been added to the review.

  1. Line 136- please explain in detail the role of STAT family of proteins in progression and metastasis along with the explicit signaling.

We added a general comment about the role of STAT in cancer progression and metastasis. Although however, in this review, we tried to focus on the part of STAT in PCa. The more detailed information about the role of the STAT family is described in the individual sections.

  1. Table 4- Please include additional details as no of patients and the competent reference.

The number of patients and, if applicable, the references have been added.

  1. Authors should include a figure illustrating the role of STAT isoforms in prostate cancer.

We want to thank the reviewer for their suggestion. We agree that the role of the isoforms would be exciting and would add impact to the reviewer. However, little could be found by the authors about STAT isoforms in PCa. Therefore, the available information has been added under the individual sections.

  1. While describing the trials in the text, please indicate the no of patients.

The number of patients has been added to the text.

  1. Line 221 , 317- please describe in detail. What kind of in vitro study? Which cell lines/model system was used?

Information about methods and cell lines have been added to the manuscript

  1. Line 323- indicate miRNAs. Ina addition, also evaluate the role of lncRNAs.

Interaction between lncRNAs and the JAK/STAT pathway is an exciting field. Our literature research revealed that several studies in the colorectal cancer cell (PMID: 31718693), inflamatory responses (PMID: 32982175), and asthma Inflammation (PMID: 33013382, PMID: 30654703) had been published. However, to our knowledge, nothing is known in PCa about the role of lncRNAs on JAK/STAT signaling so far. As this review focused on PCa, nothing about lncRNAs and JAK/STAT has been added to our manuscript.

  1. Discuss the role of STAT signaling in stem cells as the later are known for resistance and recurrence.

We added a section about STATs in CSC to the review.

  1. Please also discuss the cross talk of STAT with other pathways and NFKB contributing to prostate cancer metastasis and resistance.

A general section about the cross-talk of STAT to other pathways in prostate cancer has been added. Moreover, some general information about NFKB has been added.

  1. Reference 69- please describe the mechanism in detail and this seems important study.

We added some details about this study into the manuscript. Radioresistance has also been revealed in different other studies. However, the exact mechanism of how STAT1 mediates radioresistance in prostate cancer is not revealed yet.

  1. Authors should include 'future directions' section.

A future direction section has been added.

Round 2

Reviewer 2 Report

I congratulate the authors for the modifications. The manuscript is in much better form now. However, there are few things to be taken care of:

  1. Each subsection should end with a conclusive statement. For instance, in the newly added section of stem cells, the last line should be a summary statement of what authors draw from cited literature. In similar way, please revisit each section. Readers should have some take home message and it would give a decent impression.
  2. Table 1: does it come from patients? If yes, please indicate the no of patients. It would be valuable information. If not, please indicate the experimental model.
  3. Table 2: few trials are not annotated with references. Is it that no references available?
  4. Wherever no of patients are mentioned, please also mention the statistical inference as HR and P-value for a better impression.

Author Response

On behalf of all authors, we would like to thank the reviewer for his time and comments. Below you find the detailed response to the reviewers' comments:

  1. Each subsection should end with a conclusive statement. For instance, in the newly added section of stem cells, the last line should be a summary statement of what authors draw from cited literature. In similar way, please revisit each section. Readers should have some take home message and it would give a decent impression.

A conclusive statement has been added to each subsection.

  1. Table 1: does it come from patients? If yes, please indicate the no of patients. It would be valuable information. If not, please indicate the experimental model.

We added the experimental settings to the table. Therefore, the experimental model is now indicated.

  1. Table 2: few trials are not annotated with references. Is it that no references available?

Sadly not all studies have been published, and the information about the studies has been identified from https://clinicaltrials.gov/. Not available data has been labeled with N/A: not applicable

  1. Wherever no of patients are mentioned, please also mention the statistical inference as HR and P-value for a better impression

Due to the stage or the negative outcome of the trials, no HR and p-values are available. Therefore, the one available HR and p Values has been added to the text of the manuscript.